# Immunogenicity and Protection against Foot-and-Mouth Disease Virus in Swine Intradermally Vaccinated with a Bivalent Vaccine of Foot-and-Mouth Disease Virus Type O and A

**DOI:** 10.3390/vaccines11040815

**Published:** 2023-04-07

**Authors:** Dong-Wan Kim, Giyoun Cho, Hyejin Kim, Gyeongmin Lee, Tae-Gwan Lim, Ho-Young Kwak, Jong-Hyeon Park, Sung-Han Park

**Affiliations:** Center for Foot-and-Mouth Disease Vaccine Research, Animal and Plant Quarantine Agency, 177 Hyeoksin 8-ro, Gyeongsangbuk-do, Gimcheon-si 39660, Republic of Korea

**Keywords:** foot-and-mouth disease, intradermal vaccination, type A, type O

## Abstract

Following the worst outbreak of foot-and-mouth disease (FMD), a highly contagious disease in cloven-hoofed animals caused by the FMD virus, from November 2010–April 2011, the Korean government enforced a mandatory vaccination policy. A bivalent (FMD type O and A; O + A) vaccine has been recently implemented. Although the FMD outbreak was suppressed by vaccination, the intramuscular (IM) injection presents side effects. Therefore, improving FMD vaccine quality is necessary. Here, we investigated the side effects and immune efficacy of the O + A bivalent vaccine using two different routes of administration: intradermal (ID) and IM. To compare the immune efficacy of the two inoculation routes, virus neutralization titers and structural protein (antigen) levels were measured. The protective efficacy of ID vaccines was confirmed using two viruses (FMDV O/AS/SKR/2019 and A/GP/SKR/2018) isolated in the Republic of Korea. Serological analysis revealed that both animals administered by ID and IM injections exhibited equal immune efficacy. A virus challenge test in the target animal (swine) revealed no (or extremely low) clinical symptoms. Swine in the ID injected group exhibited no side effects. In conclusion, we suggest that the ID route of vaccination is an effective alternative to the existing IM route, which is associated with more frequent side effects.

## 1. Introduction

Foot-and-mouth disease (FMD), caused by the FMD virus (FMDV), is a highly contagious disease of cloven-hoofed animals, including swine, cattle, sheep, and goats [1]. The clinical symptoms of FMD are fever, vesicles on the foot and tongue, and impaired walking ability [1,2]. FMDV belongs to the genus *Apthovirus* within the family *Piocornaviridae* and is a positive-sense single-stranded RNA virus [1,2] with seven serological types: A, O, C, Asia1, South African Territories (SAT)1, SAT2, and SAT3 [2]. There is no effective cross-protection between different serotypes, which complicates the prevention and control of FMD [3].

In the Republic of Korea, small-scale outbreaks of FMDV occurred in April 2010 (serotype O) and January 2010 (serotype A) [4,5]. In addition, a widespread serotype O outbreak occurred in cattle and swine between November 2010–April 2011 [5]. The Korean government has implemented a mandatory vaccination policy for cattle, swine, and goats to control FMD [6]. The current vaccination program was adopted after an outbreak of FMDV serotypes A in swine in October 2018. All susceptible livestock were recently vaccinated with oil-adjuvanted inactivated bivalent vaccines containing serotype O and A antigens (O + A) [7].

During vaccination, FMD vaccine formulations containing oil-based adjuvants are repeatedly administered into the muscles of swine using an injection needle. Such intramuscular (IM) injections can cause side effects, such as granuloma, abscess, necrosis, and fibrosis in the injection area. In the Republic of Korea, these side effects cause economic losses to meat producers who discard the deformed shoulder region of the pork, which is more expensive than the other parts [8]. Several studies on other vaccines have investigated the effects of vaccine injection through the intradermal (ID) route [9,10,11]. ID injections that induce an efficient immunological response against FMD and confer protective immunity can serve as a good alternative for IM injections [12,13]. In addition, IM injections are most commonly recommended to be delivered in doses of 2 mL, whereas the dosage required for ID injections is lower. Therefore, ID injection can reduce the production cost per dose of the FMD vaccine [12]. Accordingly, establishing effective vaccination strategies against FMDV is important depending on the country’s situation.

In this study, we injected the bivalent O + A vaccine through the IM and ID routes and evaluated the differences in immune efficacy and side effects. We also showed that when injected through the ID route, the bivalent vaccine is effective against two types of FMDVs—O/AS/SKR/2019 and A/GP/SKR/2018—isolated in the Republic of Korea.

## 2. Materials and Methods

### 2.1. Cells

A suspension of the baby hamster kidney (BHK-21) cells were provided from the American Type Culture Collection (CCL-10). This cell lines were used for virus passage and FMD vaccine antigen production. The cell line was adapted to grow in the serum-free Cellvento BHK200 cell culture medium (Merck, Darmstadt, Germany) incubated in a shaking incubator (110 rpm, 37 °C, and 5% CO_2_). Fetal porcine kidney (LF-BK) cells were provided from Plum Island Animal Disease Center (Greenport, NY, USA) and used for produce FMDV strains and for the virus neutralization (VN) test. LF-BK cells were cultured in Dulbecco’s modified Eagle’s medium (Corning, NY, USA) with 10% fetal bovine serum (FBS; Gibco, NY, USA) and 1% penicillin–streptomycin (P/S; Gibco, NY, USA) during maintenance, and with 2% FBS during FMDV infection at 37 °C with 5% CO_2_.

### 2.2. Viruses

The O PanAsia-2 (O PA-2) and A_22_ Iraq/24/64 (A_22_ Iraq, GenBank accession number AY593764.1) viruses were adapted to BHK-21 suspension cells. The other viruses used for the VN test were O_1_ Manisa/Turkey/69 (O_1_ Manisa), O/Boeun/SKR/2017 (O BE, GenBank accession number MG983730.1), O/Andong/SKR/2010 (O AD, GenBank accession number KC503937), O/Jincheon/SKR/2014 (O JC, GenBank accession number 162590.1), O/Gimje/SKR/2016 (O GJ, GenBank accession number KY086465.1), O/Anseong/SKR/2019 (O AS, GenBank accession number KU991734.1), O/VIT/2013 (GenBank accession number KY492067.1), Taiwan 97 (O TWN 97, GenBank accession number KJ831708.1), A/Yeoncheon/SKR/2017 (A YC, GenBank accession number KY766148.1), A/Pocheon/SKR/2010 (A PC, GenBank accession number KC588943.1), A/Gimpo/SKR/2018 (A GP, GenBank accession number MK463492.1), and A/Nepal/12/2017 (A NEP).

### 2.3. Production of FMD Vaccine Antigens

The multivalent FMD vaccine used in this study contained the O PA-2 and A_22_ Iraq (O PA-2 + A_22_ Iraq) strains. Both strains were inoculated into BHK-21 suspension cells (multiplicity of infection, 0.005) and incubated in a shaking incubator (110 rpm, 37 °C, and 5% CO_2_). The viruses were harvested at 16 h post-infection (hpi) and centrifugated at 1760× *g* for 20 min at 4 °C to remove cell debris. To inactivate the clarified viruses, 3 mM binary-ethylenimine (Sigma-Aldrich, St. Louis, MO, USA) was added, and the mixture was incubated in a shaking incubator for 24 h at 26 °C [14]. The inactivated viruses were treated with a final concentration of 7.5% polyethylene glycol 6000 (Sigma-Aldrich) and 0.5 M NaCl (Sigma-Aldrich) for 16 h at 4 °C. The mixture was centrifugated at 10,000× *g* for 30 min at 4 °C, and the precipitate was purified by the sucrose density gradient ultra-centrifugation method as described in a previous study [15,16]. The vaccine was manufactured to contain 15 µg/dose each of the O PA-2 and A_22_ Iraq antigens and was supplemented with 50% adjuvant (ISA 207; SEPPIC, Paris, France) according to the manufacturer’s recommendation for swine. The ISA 207 adjuvant has been manufactured for specific efficacy in ID vaccines. Following this, 1% saponin (Quil-A) and 10% aluminum hydroxide (Al(OH)3) were added to the vaccine to enhance its immunogenicity.

### 2.4. Vaccination and Challenge

The animal experiments were conducted with the approval of the Institutional Animal Care and Use Committee of the Animal and Plant Quarantine Agency of Republic of Korea (IACUC approval number 2020-506). First, antibody titers were tested in guinea pigs to compare the immunogenicity of ID and IM. Six-week-old female Hartley guinea pigs were divided into groups (ID and IM, n = 5) and inoculated with the corresponding vaccine (one-fifth dose). A booster vaccine (second dose) was administered 4 weeks after the first inoculation. Blood samples were collected at 0, 7, 14, 21, 28, and 35 d post-vaccination (dpv) (Table 1). Next step, immunogenicity was tested in a target animal, swine. The swine in the ID group were vaccinated through the ID route (0.5 mL vaccine/dose). ID vaccination was performed using Pulse 250 needles (Pulse Needle Free systems, Lenexa, KS, USA). The immunogenicity of 8–10-week-old farm pigs (n = 5 in each group) was confirmed similarly, changing only the volume of vaccine (1 dose) (Table 1).

This study was conducted in an Animal Biosafety Level 3 facility at the Animal and Plant Quarantine Agency (APQA). Swine were challenged with 3 weeks after the FMD ID vaccination according to the guidelines of the Office International des Epizooties (OIE). The challenge was conducted using O AS and A GP strain of FMDV, and there were three groups: negative (unvaccinated, n = 3), donor (contact infection, n = 3), and ID vaccination (n = 5). FMDV were inoculated with 1 × 10^6^ tissue culture infectious dose (TCID)/mL (volume, 0.1 mL) into the hoof of donor swine. The donor swine was allowed to coexist with other groups to induce contact infection. The study lasted 7 d, and donors were sacrificed 3 d post-challenged (dpc). Clinical symptoms and body temperature of the swine were monitored daily. For clinical symptoms, the counting standard included eight points: one point for each of the three hooves, the tongue, mouth, nose, limping, and decreased appetite. Swab and serum samples of the experimental swine were obtained at 0, 3, 5, and 7 dpc (Table 2).

### 2.5. Real-Time Reverse Transcription Polymerase Chain Reaction (RT-PCR)

Viral RNA was extracted from the serum samples of experimental animals using the Maxwell^®^ RSC RNA extraction kit (Promega Corporation, Madison, WI, USA) following the manufacturer’s instructions. Real-time RT-PCR was conducted using a one-step prime-script RT-PCR kit (Bioneer, Daejeon, Republic of Korea) according to the manufacturer’s protocol. The primers used for RT-PCR targeted the FMDV 3D region (5′-GGA ACY GGG TTT TAY AAA CCT GTR AT-3′ and 5′-CCT CTC CTT TGC ACG CCG TGG GA-3′). The probe (5′-CCC ADC GCA GGT AAA GYG ATC TGT A-3′) was labeled with 6-FAM and TAMRA. PCR was performed using the CFX 96 Touch™ Real-Time PCR Detection System (Bio-Rad Laboratories, Hercules, CA, USA).

### 2.6. Enzyme-Linked Immunosorbent Assay (ELISA)

The PrioCHECK FMDV Type O Antibody ELISA Kit (Thermo Fisher Scientific, Cleveland, OH, USA) and VDPro FMDV Type A AB ELISA kit (MEDIAN Diagnostics, Chuncheon-si, Republic of Korea) were used to confirm the presence of FMDV type O or A antibodies in the inactivated serum samples of experimental animals. Positive values were confirmed if the percentage inhibition (PI) was >50% (type O kit) or if the optical density was ≥0.4 (type A kit).

### 2.7. Viral Neutralization Test (VNT)

VNT was performed according to the methods described in the OIE terrestrial manual [17]. Serially diluted serum samples were reacted with 200 TCID50/0.05 mL of FMDV type O or A and incubated at 37 °C for 1 h. After the reaction, the viruses were inoculated into LF-BK cells and incubated at 37 °C for 2–3 d [17]. Antibody titers were calculated using the Spearman–Kärber method [18].

### 2.8. Statistical Analysis

All data are presented as the mean ± standard error of the mean (SEM). Individual variances were computed for each comparison, and statistical analyses were performed using two-way analysis of variance (ANOVA) followed by Tukey’s multiple comparisons test in the Prism software (version 9; GraphPad Software, San Diego, CA, USA). *, *p* < 0.05 (significant); **, *p* < 0.01 (very significant); ***, *p* < 0.001 (highly significant); ****, *p* < 0.0001 (extremely significant); and ns, *p* > 0.05 (not significant).

## 3. Results

### 3.1. Immunogenicity of ID Injection Using the FMDV O + A Bivalent Vaccine

Using the two vaccine strains (O PA-2 and A_22_ Iraq), VN titers were determined against FMDVs that occurred in the Republic of Korea and neighboring countries. Eight-week-old swine were vaccinated, and serum samples obtained at 35 dpv were used in the VNT. Among swine vaccinated against FMDV type O, VN titers were higher than 1:300 against nine viruses (except the O JC strain, which showed a VN titer of 1:100; Figure 1a). FMDV type A is a domestically occurring virus, and swine vaccinated against this strain showed VN titers ranging from 1:100 to 1:300 (Figure 1b). Immunogenicity could not be confirmed against the A NEP strain of FMDV.

In guinea pigs, the results of type O structural protein (SP)-ELISA showed that both the IM and ID groups, the average PI value was 50% at 21 dpv and >80% at 35 dpv (Figure 2a). At 21 dpv, both groups (IM and ID) showed VN titers of ≥1:100 in experiments evaluating the neutralizing antibody titer against the homologous type O PA-2 strain (Figure 2b). The results of type A SP-ELISA were positive in both groups (IM and ID) at 28 dpv; the VN titers were lower than those of type O but showed a similar tendency (Figure 2c,d). 

In swine vaccination experiment, the IM and ID groups showed positive values after 21 dpv in both type O and type A SP-ELISA (Figure 3a,c). This pattern was similar to that observed in the guinea pig experiment, and the positivity rate was 70–80% at the end of the experiment. In swine, the VN titer was >1:30 in the ID group at 7 dpv (the initial stage of vaccination; Figure 3b,d). However, both groups (IM and ID) showed similar immunogenicity after 21 dpv. At the end of the experiment, the VN titers were >1:300 in both groups.

### 3.2. Efficacy of the FMDV Strain O/AS/SKR/2019 Vaccine

#### 3.2.1. Serological Responses in Swine after Immunization

Both types of SP-ELISA (types O and A) showed a PI value of 40% at 14 dpv and a positive value at the end of the experiment (Figure 4a,b). Average VN titers were ≥1:30 at 14 dpv and ≥1:100 after a challenge with the O AS strain. At the end of the experiment, the VN titer was approximately 1:300 (Figure 4c).

#### 3.2.2. Detection of FMDV in Virus-Challenged Swine by Real-Time RT-PCR

Of the five swine in the ID vaccinated group, one individual showed sporadic viremia lasting 3 dpc, with virus copy numbers (log 10) ranging from 0.115 to 1.215. The virus copy number (log 10) was 3.256–5.152 in the contact infection group. Individuals in the contact infection group were sacrificed after 3 dpc. In the negative control group, viral RNA was detectable for 2–7 dpc, with viral copy numbers (log 10) ranging from 1.752 to 7.256 (Figure 4d).

#### 3.2.3. Clinical Symptoms in Virus-Challenged Swine

We checked for viral infections in swine by inspecting the hoof, mouth, and tongue (non-challenged regions) for the presence of vesicles. In the contact infection group, an average of 5 points were measured at 3 dpc, after which the individuals were sacrificed. Clinical symptoms were confirmed in the negative control group at 2 dpc, and the total score was calculated using 7 points at 5 dpc (Figure 4e). There were no symptoms in the ID injection group.

### 3.3. Efficacy of the FMDV Strain A/GP/SKR/2018 Vaccine

#### 3.3.1. Serological Responses in Swine after Immunization

In the ID group, type O SP-ELISA showed a PI value of 40% at 7 dpv (Figure 5a). In addition, the positive value was maintained until the end of the experiment, indicating the high immunogenicity of the vaccine. Immunogenicity was slightly lower in the type A SP-ELISA but showed a similar tendency (Figure 5b). The VN titer was ≥1:10 in 75% of vaccinated swine (3/4) at 14 dpv. From 21 to 28 dpv, the average VN titer of vaccinated swine was >1:45 (Figure 5c).

#### 3.3.2. Detection of FMDV in Virus-Challenged Swine by Real-Time RT-PCR

The ID vaccinated group showed sporadic viremia lasting 3 dpc, with virus copy numbers (log 10) ranging from 0.028 to 1.487. The virus copies numbers (log 10) ranged from 3.668 to 4.865 in the contact infection group, and the individuals were sacrificed after 3 dpc. In the negative control group, viral RNA was detectable during 2–7 dpc, with virus copy numbers (log 10) ranging from 1.654 to 7.827 (Figure 5d).

#### 3.3.3. Clinical Symptoms in Virus-Challenged Swine

We checked for viral infections by inspecting the swine’s hoof, mouth, and tongue (non-challenged regions) for the presence of vesicles. In the contact infection group, an average of 3 points were measured at 3 dpc, following which the individuals were sacrificed. Clinical symptoms were confirmed in the negative control group at 3 dpc, and the score was calculated using 7 points at 5 dpc (Figure 5e). In the ID group, one out of five swine showed slight symptoms. Temperature is considered a factor when confirming clinical symptoms, and temperatures > 40 °C are confirmed with 1 point. One pig showing symptoms was confirmed to have a body temperature ≥ 40 °C or higher, but no other clinical symptoms were noted.

### 3.4. Confirmation of Differences in the Occurrence of Side Effects According to the Injection Route of the FMD Vaccine

Side effects were recorded and analyzed in swine from both groups (IM and ID) used for comparisons of immunogenicity. The side effects were confirmed by dissecting the FMD vaccination site (the neck behind the ear). No side effects were found among five swine in the ID group (Figure 6). In the IM group, side effects were confirmed at the inoculation site in three out of five swine as expected due to vaccination.

## 4. Discussion

FMD is endemic to Southeast and East Asia, with multiple serotypes (and multiple genotypes in each serotype) under circulation. Although some countries (such as the Republic of Korea and Japan) were initially free of FMD, FMD outbreaks were recorded in 2010. Since these initial outbreaks, the Republic of Korea has experienced several FMD outbreaks despite regular vaccination programs [4,5]. Vaccination using high antigen payload vaccines is an important method of controlling FMDV occurrence and spread, as this limits the possibility of transmission and minimizes the duration and intensity of outbreaks.

ID injection targets the dermal and epidermal layers, rich sources of antigen-presenting cells (such as Langerhans cells, dermal dendritic cells, and dermal macrophages) and are known to produce vaccine-induced immune responses [19]. Due to these advantageous features, several studies have investigated ID vaccines [8,9,10]. A previous study on a needle-free delivery system reported that FMD vaccination through the ID route using the same antigen as that used for IM vaccination effectively protected swine from FMDV [20].

Existing IM FMD vaccines using oil-based adjuvants can cause fatal side effects, such as necrosis at the inoculation site. However, the ID vaccine spreads intradermally and is expected to reduce the occurrence of side effects associated with IM vaccines that spread intramuscularly. We found no side effects in the ID injection group in this study. We confirmed the immune efficacy of the ID vaccine with 0.5 mL/dose, which is one-fourth the amount used for the IM vaccine. The two inoculation routes exhibited equivalent effects on immunity, as confirmed using SP-ELISA and VNT. A safety test was conducted by 3-fold dose ID vaccination in guinea pigs, and all individuals survived to ensure safety. Additionally, stability was tested by comparing immunogenicity 0, 3, and 6 months after ID vaccine production (data not shown).

The vaccine developed in this study conferred protection against the FMDV involved in a recent outbreak in Korea. The vaccine against FMDV type O AS conferred 100% protection. When vaccinated against FMDV type A GP, one swine (out of five) showed slight symptoms. Guidelines regarding the clinical symptoms of FMD [21] suggest that a body temperature of ≥40 °C is considered a symptom. In addition to temperature, slight symptoms were observed at the site of inoculation in the one pig tested. Unlike type O FMDV, type A has a very narrow protective spectrum. As a result, the rate of vaccine matching is very low. These issues require more research, and further studies should investigate whether additional steps (such as increasing the amount of antigen in the vaccine or adding an adjuvant) can increase the immunogenicity of the vaccine.

Since FMD vaccination started in 2011 in the Republic of Korea, one animal has been recommended to be vaccinated with one needle. However, this practice is not well maintained in the field. Using the same needle multiple times for multiple animals can lead to secondary infections through the needle [22]. A study reported that economic losses in the pig industry increased significantly after FMD vaccination [8,23]. In addition, intramuscular residual vaccines were found to be the main cause of chronic inflammation [22]. ID injection using a needle-free system can minimize this a potential disadvantage of ID vaccination [24]. However, such needle-free systems are in the early stages of development and have certain limitations. For example, a needle-free system may cause exhaustion in the wrist due to its heavy weight and the method of inoculation. More research on needle-free systems is needed to develop a quick, simple, reliable vaccination technique.

## 5. Conclusions

In conclusion, we found that the economic losses caused by the side effects of IM injection can be greatly reduced by using the ID route for vaccination in the pig industry. The vaccines developed here showed similar immune efficacy, indicating that novel vaccines can be a better alternative to the existing FMD vaccine. However, more research on ID FMD vaccine compositions is needed to ensure that ID vaccines can confer higher immunity levels. In addition, further studies should investigate methods for early protection, which is an advantage of the ID vaccination route.

## Figures and Tables

**Figure 1 vaccines-11-00815-f001:**
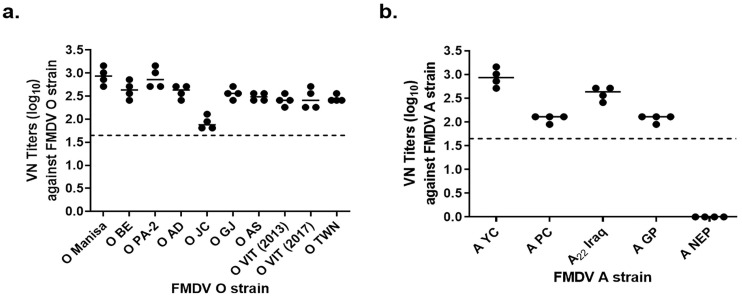
Neutralizing antibodies against various FMDV strains induced in intradermally vaccinated swine. The test vaccine was administered to swine at 15 µg O PA-2 + A22 Iraq antigen, ISA207 (oil-based emulsion, 50%, W/O/W), 10% Al(OH)3, and 1% Saponin. (**a**) Viral Neutralization (VN) titers against FMDV O strains; (**b**) VN titers against FMDV A strains.

**Figure 2 vaccines-11-00815-f002:**
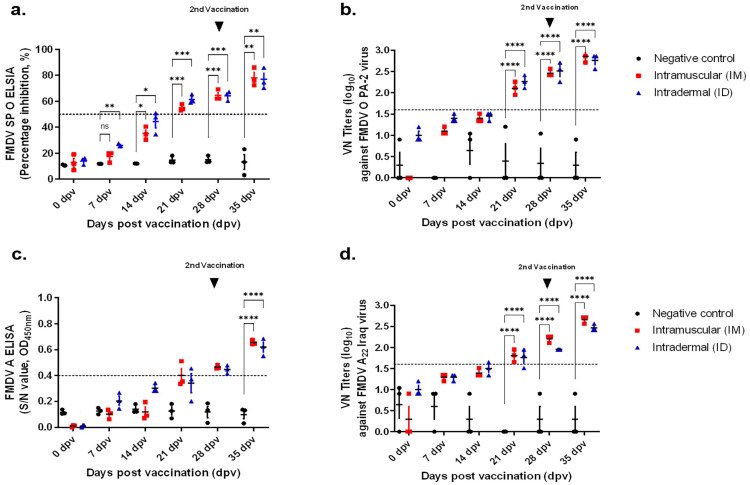
Comparison of serological response after vaccination of guinea pigs with ID and IM vaccine. The test vaccine was administered to guinea pigs at one-tenth dose of O PA-2 + A22 Iraq antigen for pig use, ISA207 (oil-based emulsion, 50%, W/O/W), 10% Al(OH)3, and 1% Saponin. The negative control group was administered the test without vaccine. The IM group was inoculated with 0.2 mL/dose of vaccine and ID group was inoculated with 0.05 mL/dose of vaccine. Serum sampled from guinea pigs at 0, 7, 14, 21, 28, and 35 dpv was analyzed via SP antibody ELISA and VN assay. (**a**) Percentage inhibition (PI) value of FMDV type O SP ELISA in individuals (The dashed lines are positive value.). (**b**) VN titers against the FMDV O PA 2 strain. (**c**) Percentage inhibition (PI) value of FMDV type A SP ELISA in individuals (The dashed lines are positive value.). (**d**) VN titers against FMDV A22 Iraq strain. The dotted lines in SP-ELISA results indicate 50% inhibition, the positive threshold in the test. The dotted lines in VN test show 1.65 log VN titers. IM: intramuscular injection; ID: intradermal injection; the datasheets are the mean ± SEM; statistical analyses were performed using two-way ANOVA; ns; *, *p* < 0.05; **, *p* < 0.01; ***, *p* < 0.001; and ****, *p* < 0.0001.

**Figure 3 vaccines-11-00815-f003:**
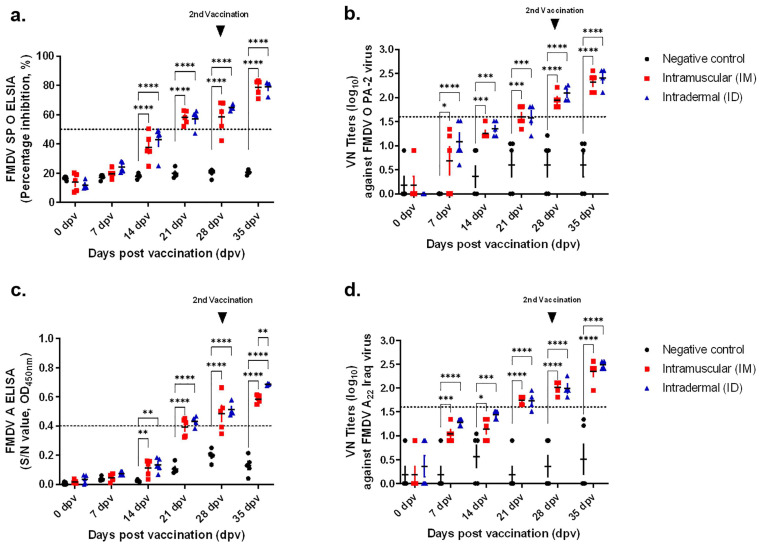
Comparison of serological response after vaccination of swine with IM and ID vaccine. The test vaccine was administered to swine at 15 µg O PA-2 + A22 Iraq antigen, ISA207 (oil-based emulsion, 50%, W/O/W), 10% Al(OH)3, and 1% Saponin. A negative control group was administered the test without vaccine. The IM group was inoculated with 0.2 mL/dose of vaccine and ID group was inoculated with 0.5 mL/dose of vaccine. Serum sampled from swine at 0, 7, 14, 21, 28, and 35 dpv was analyzed via SP ELISA and VN test. (**a**) Percentage inhibition (PI) value of FMDV type O SP ELISA in individuals. (**b**) VN titers against O-PA2 strain. (**c**) Percentage inhibition (PI) value of FMDV type A SP ELISA in individuals. (**d**) VN titers against A22 Iraq strain. IM: intramuscular injection; ID: intradermal injection; the datasheets are the mean ± SEM; statistical analyses were performed using two-way ANOVA; *, *p* < 0.05; **, *p* < 0.01; ***, *p* < 0.001; and ****, *p* < 0.0001.

**Figure 4 vaccines-11-00815-f004:**
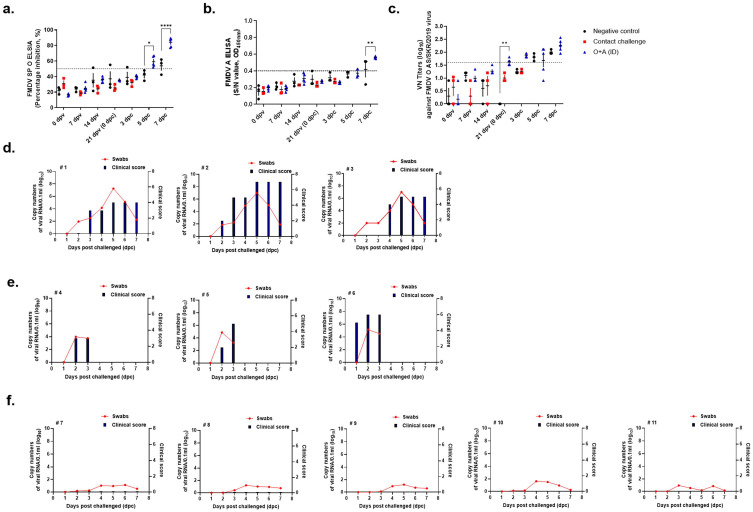
Clinical scores and FMDV RNA levels in swine immunized and VN titer after contact infection with donor swine inoculated with the O/AS/SKR/2019 strain. After vaccination, the swine were challenged with the O/AS/SKR/2019 strain. FMDV RNA levels in nasal swabs were measured by qRT-PCR from 0 to 7 days after challenge. Serum samples from the swine at 0, 7, 14, and 21 dpv, and 0, 3, 5, and 7 dpc were analyzed via VN assay. (**a**) Percentage inhibition (PI) value of FMDV type O SP ELISA in individuals. (**b**) Percentage inhibition (PI) value of FMDV type A SP ELISA in individuals. (**c**) VN titers against O/AS/SKR/2019 strain. (**d**) The clinical scores and FMDV RNA levels in negative group. (**e**) The clinical scores and FMDV RNA levels in contact infection group. (**f**) The clinical scores and FMD virus RNA levels in ID vaccination group. ID: intradermal injection; the datasheets are the mean ± SEM; statistical analyses were performed using two-way ANOVA; *, *p* < 0.05; **, *p* < 0.01; ***, *p* < 0.001; and ****, *p* < 0.0001.

**Figure 5 vaccines-11-00815-f005:**
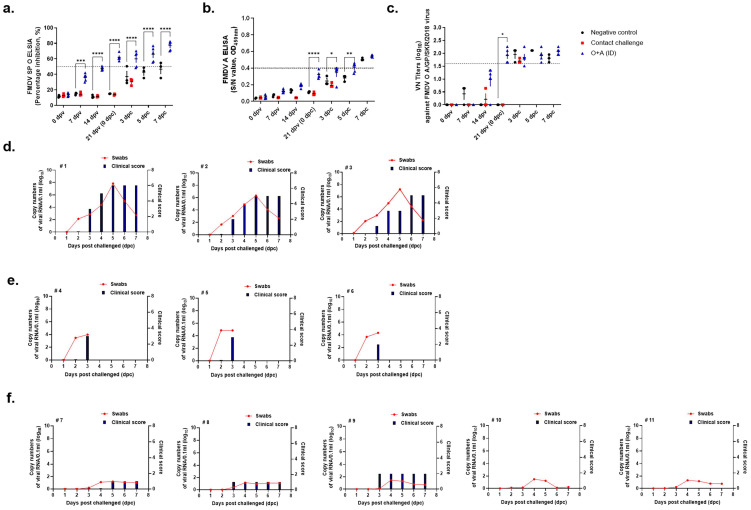
Clinical scores and FMDV RNA levels in swine immunized and VN titer after contact infection with donor swine inoculated with the A/GP/SKR/2018 strain. After vaccination, the swine were challenged with the A/GP/SKR/2018 strain. FMD virus RNA levels in nasal swabs were measured by qRT-PCR from 0 to 7 days after challenge. Serum samples from the swine at 0, 7, 14, and 21 dpv, and 0, 3, 5, and 7 dpc were analyzed via VN test. (**a**) Percentage inhibition (PI) value of type O SP ELISA in individuals. (**b**) Percentage inhibition (PI) value of type A SP ELISA in individuals. (**c**) VN titers against O/SA/SKR/2019 strain. (**d**) The clinical scores and FMD virus RNA levels in negative group. (**e**) The clinical scores and FMD virus RNA levels in contact infection group. (**f**) The clinical scores and FMD virus RNA levels in ID vaccination group. ID: intradermal injection; the datasheets are the mean ± SEM; statistical analyses were performed using two-way ANOVA; *, *p* < 0.05; **, *p* < 0.01; ***, *p* < 0.001; and ****, *p* < 0.0001.

**Figure 6 vaccines-11-00815-f006:**
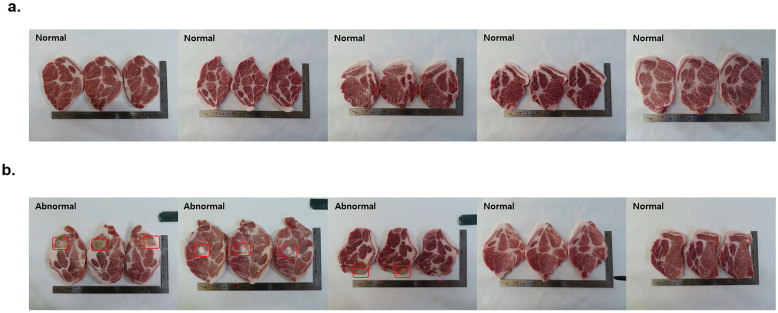
Pathologic lesion of an injection site following administration of the ID and IM bivalent vaccine (O PA-2 + A22 Iraq). After vaccination of five swine with ID and IM bivalent vaccine, the pathologic lesion of injection sites was checked. (**a**) The ID vaccination group; (**b**) The IM vaccination group (The marked red boxes are the part of the side effects.).

**Table 1 vaccines-11-00815-t001:** Experimental designs of vaccination methods.

Group	No. of Guinea Pigs	Vaccine Strain	Serum Obtained at Dpv
Negative (unvaccinated)	3	-	0, 7, 14, 21, 28, 35
IM	5	O PA-2 + A22	
ID	5		
**Group**	**No. of swine**	**Vaccine strain**	**Serum obtained at dpv**
Negative (unvaccinated)	5	-	0, 7, 14, 21, 28, 35
IM	5	O PA-2 + A22	
ID	5		

dpv, day post-vaccination; ID, intradermal; IM, intramuscular.

**Table 2 vaccines-11-00815-t002:** Experimental designs of intradermal vaccination against FMD virus.

Group	No. of Swine	Vaccine Strain	Challenge Virus	Serum Obtained at Dpc	Swab Obtained at Dpc
Negative (unvaccinated)	3	-	-	−21, −14, −7, 0, 3, 5, 7	1–7
Donor (contact infection)	3	-	O/AS/SKR/2019 or A/GP/SKR/2018	−21, −14, −7, 0, 3	1–3
ID	5	O PA-2 + A22	-	−21, −14, −7, 0, 3, 5, 7	1–7

dpc, day post-challenge; ID, intradermal.

## Data Availability

Not applicable.

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
