# Peer review of "Immunogenicity and Protection against Foot-and-Mouth Disease Virus in Swine Intradermally Vaccinated with a Bivalent Vaccine of Foot-and-Mouth Disease Virus Type O and A"

_vaccines, 2023, doi:10.3390/vaccines11040815_

Round 1
Reviewer 1 Report
The manuscript compared immunogenicity and protection against foot and mouth disease virus (FMDV) in swine vaccinated with a bivalent vaccine of type O and A by the routes of intramuscular (IM) and intradermal (ID). Both inoculation routes provided well immune and protection efficacy in swine, while the ID injected pigs showed no side effects compared with the IM injection. It maybe provides an effective alternative vaccinated route for FMDV vaccination, but has less interest for FMDV vaccine development. Therefore, it is not publishable in Vaccine in current version.
1. Line 43-44, “All susceptible livestock were recently vaccinated with oil-adjuvant inactivated bivalent vaccines containing serotype O and A antigens (O+A)”. The authors should explain why the experiment did not use the commercial vaccine to compare the IM and ID vaccination routes. Otherwise, the commercial vaccine control should be added in the experiment.
2. In the 3.2 and 3.3 Result sections, the description and data of antibody level, virus RNA, and clinical symptoms in the IM group should be provided to compare with that of ID group.
Author Response
I sincerely appreciate your comments.
The answer to your feedback will be sent as an attachment.
Please see the attachment.
I hope you are having a great day.

Reviewer 2 Report
The objective of this study was to assess the immunity and protection efficacy of a bivalent inactivated and adjuvanted FMDV vaccine when delivered intradermally (ID) based on comparison with current intramuscular (IM) injection method.
The main findings from this study is that ID injection of the FMDV vaccine had no side effect such as inflammation and granuloma formation in vaccine injected neck area, but induced comparable levels of protective immunity and clinical protection in a challenge study. The authored confirmed the immune efficacy of the ID vaccine with 0.5 ml/dose, which is 1/4th the amount used for the IM vaccine.
Overall well-written manuscript, but some expressions used in this manuscript need to be rephrased into more clear and succinct sentences. The sentences recommending rephrasing are listed as minor comments below. This reviewer has no major comments, but following minor comments need to be addressed before further decision is made.
Minor comments:
Line 18: ‘using two different routes’ instead of ‘according to the route’
Line 22: add ‘by’ after administered.
Line 44: oil-adjuvanted (add ed)
Line 53: protective instead of anti-infectious
Line 107: after the first inoculation instead of after inoculation
Line 113: Experimental design and vaccination methods looks better title for Table 1. Negative control looks unclear. Is this group ‘unvaccinated and challenged group’? The reviewer recommends making it clear in the table, figure legends and in other sections.
Line 129: vaccination instead of vaccine in Table 2 title
Line 163: determined instead of confirmed.
Line n169: ‘Immunogenicity was not tested’ instead of ‘could not be confirmed’
Line 185: ‘Neutralizing antibodies against various FMDV strains induced in intradermally vaccinated swine’ looks better for Figure 1 title.
Line 175: ‘both’ instead of ‘in’
Lines 178 and 180: Figure numbers look not correct. Need corrections.
Lines 182 and 183: Unclear sentence. Rephrase to make it clear.
Line 189: vaccine antigen instead of ‘vaccine’.
Line 199: ‘In swine vaccination experiment’ instead of ‘Among swine’
Line 200: Figure 3b looks not correct. 3C looks correct.
Line 203: 3b instead of 3c.
Line 292: ‘five swine as expected due to vaccination’ instead of ‘five swine and were expected to have been caused by FMD vaccination
Line 295: injection sites instead of ‘an injection site’
Line 304: programs instead of drives
Line 323: ‘equal to positive value’ is not clear to understand. Make it clearer.
Line 326: In addition to instead of ‘Apart from’
Line 328: protective instead of ‘defense’
Line 329: further studies instead of ‘future studies’
Line 348: further instead of future
Line 349: do you mean ‘disadvantage’ in this sentence? Change it as ‘a potential disadvantage of ID vaccination’ for clarity.
Author Response
I sincerely thank you for your comments to improve the quality of our paper.
I have modified each sentence according to your comments.
The modified manuscript is also attached.
I hope you are having a great day.
Reviewer 3 Report
The authors describe the comparative performance of a bivalent aqueous-based FMDV vaccine (O PA-2 and A22 Iraq strains) administered by the IM or ID routes in guinea pigs and swine. Humoral immune responses were assessed in both species by indirect commercial ELISA or VNT against the homologous strains. The immunization schedule comprises two immunizations at 0 and 28 days after primary vaccination, and serum samples were obtained at different times post-vaccination. Protection was assessed (at least in some of the pigs) after vaccination (not clear when) by in vivo challenge against two FMDV strains (O and A serotypes), heterologous to those of the vaccine strains. A recent review on the skin-based immunity and protection elicited in pigs (https://doi.org/10.3390/vaccines11020450) revealed that around 10 publications have presented results in this area following different strategies in the last few years.
In this respect, the main novelty of this manuscript is that they use a bivalent vaccine which is tested (only for the ID-vaccinated groups) by in vivo challenge against two FMDV strains (O and A serotypes), which differed from those used in the vaccination
The organization of the text and its presentation specially for the Material and Methods and Results sections are rather confusing, including repetitions, gaps, and poor text connections between both sections. The consecution of results is somewhat messy and makes it difficult to understand the real value of the results. Some bibliographic references are repeated (# 3 and #28), and some results are missing, preventing proper comparison of the immunization protocols in the target species.
There are several points that should be addressed before the final evaluation of the manuscript.
· The nomenclature and acronyms corresponding to the FMDV strains should revised and harmonized throughout the text. For example: O/AS/SKR/2019 strain corresponds to the “O AS” and to the “O AS 2015” acronyms?
· Protocols and Methods used for some of the results presented are not included in the Material and Methods section (for example, the heterologous VNT assays in figure 1).
· Section “2.4 Vaccination and Challenge” should be rephrased for clarity (for example, is the IM dose applied to pigs similar to the ID dose?)
· No comments are made about the safety, innocuity and stability of the in-house performed vaccines
· References #8 and #23 are the same. Several references with this and other vaccines applied through this route should be included (10.3390/pathogens10091115 - 10.1016/j.virol.2016.12.021, 10.1186/1743-422X-7-215 etc), including the first ones focused on this strategy ( 10.1016/j.vaccine.2007.06.010 - 10.1016/j.vaccine.2006.09.066 - 10.1016/j.vaccine.2012.02.049 - 10.1016/j.vaccine.2008.12.011)
· Description of the challenge experiments in pig is very confusing.
· It is not clear when the challenge was performed: “…for the challenge, the O AD or A GP strains of FMDV were inoculated in combination with 1 x 10e6 tissue culture infectious dose (TCID)/ml (volume, 0.1 ml) into the hoof of each pig 3 weeks after vaccination. Swine were challenged with the FMDV 3–4 weeks after the FMD vaccination…”
· the authors inform that the challenge was performed with the O AD or A GP strains, but it is unclear if the challenge was performed by needle injection or contact. Some animals were needle-infected and others through “contact”? This should be explained in detail. The authors inform that the challenge was performed with the O AD or A GP strains, but it is not clear how the challenge was performed: needle injection or contact? This should be explained in detail.
· The experimental design includes the IM-vaccinated group, but the challenge results of this experimental group are not shown. It is important to include those results to evaluate its comparison with the ID-vaccinated group.
· Since antibody levels raised to acceptable values after 28 dpv, the inclusion of a second dose of the vaccines was due to secure a protective status of the challenged animals? What do the authors anticipate regarding the duration of the specific immunity in these animals and which are the bases of such assumption?
Author Response

(The authors gave the same response as above.)

Round 2
Reviewer 1 Report
I am sorry that the authors didn't answer my major concerns. No additional data was added to compare the vaccination effect by IM and ID injections in the revised version.
Author Response
Sorry for replying again because I didn't answer your question correctly.
Responses to comments have been uploaded as documents.
We really appreciate your comments to improve the quality of this study.

Round 3
Reviewer 1 Report
OK.